# Sub-TeV hadronic interaction model differences and their impact on air showers

**Michael Schmelling[1]⋆, Álvaro Pastor-Gutiérrez[1],
Harm Schorlemmer[1,2,3] and Robert Daniel Parsons[1,4]**

**1** Max-Planck-Institut für Kernphysik, Heidelberg, Germany
**2** IMAPP, Radboud University Nijmegen, Nijmegen, The Netherlands
**3** Nationaal Instituut voor Kernfysica en Hoge Energie Fysica (NIKHEF),
Science Park, Amsterdam, The Netherlands
**4** Institut für Physik, Humboldt-Universität zu Berlin, Berlin, Germany

⋆ michael.schmelling@mpi-hd.mpg.de

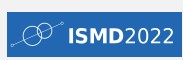

*51st International Symposium on Multiparticle Dynamics (ISMD2022)
Pitlochry, Scottish Highlands, 1-5 August 2022*

## Abstract

**In the sub-TeV regime, the most widely used hadronic interaction models disagree significantly in their predictions of particle spectra from cosmic ray induced air showers. We investigate the nature and impact of model uncertainties, focussing on air shower primaries with energies around the transition between high and low energy hadronic interaction models, where the dissimilarities are largest and which constitute the bulk of the interactions in air showers.**

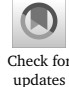

## 1 Introduction

The description of air showers created by high energy cosmic ray primaries hitting the atmosphere requires the modelling of hadronic interactions between elementary particles and air nuclei over many orders of magnitude in energy. The need to understand hadronic physics over such a large energy range is highlighted for example by the observation of an unexpectly large muon flux in high energy cosmic-ray interactions [1, 2], or in estimates of the physics reach of future astroparticle physics experiments [3]. Model comparisons focussing on primaries with energies in the range from 100 GeV to 100 TeV are discussed in [4], and a study of how model predictions compare to experimental data recorded at the LHC can be found e.g. in [5]. However, the fact that in extensive air showers the bulk of the particle production happens late in the shower evolution puts emphasis also on the low energy region below 100 GeV. This has been studied in [6], some key results of which are summarised below.

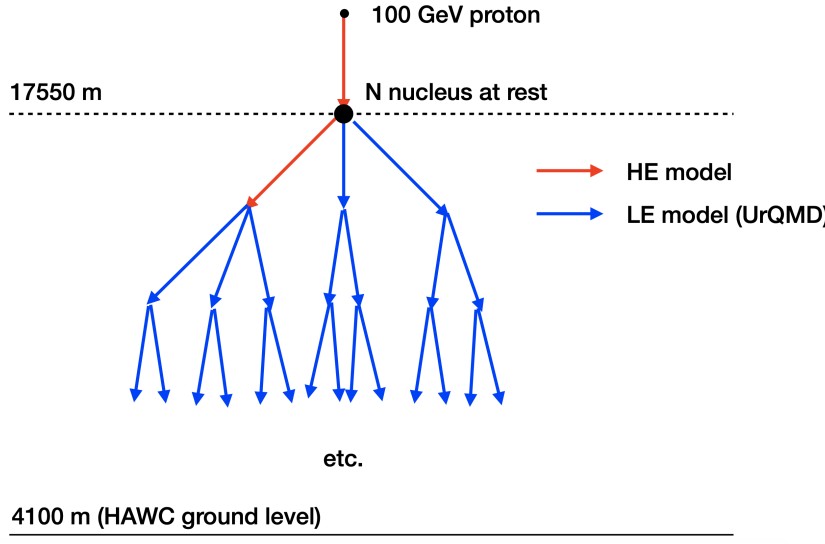

Figure 1: Sketch of the scenario considered in this study.

The scenario considered is sketched in fig. 1 and modelled by the CORSIKA v7.64 [7] air shower simulation software package. A primary proton with a total lab energy of 100 GeV and zero zenith angle interacts with a nitrogen nucleus at an altitude of 17550 m. The observation level for the final state is at 4100 m. Fixing the height of the first interaction removes geometric effects caused by varying ground distances and puts the focus on differences between the physics modelling. Here EPOS-LHC [8], QGSJetII-04 [9], SIBYLL 2.3c [10] and UrQMD [11] are considered. UrQMD is designed for lab energies from less than 100 MeV up to O(200) GeV, the other models can be applied for lab energies above O(40) GeV. A full shower simulation starts with a high-energy (HE) model for the initial part of the shower development and switches to a low-energy (LE) model at a transition energy, which in CORSIKA is set to 80 GeV. With a 100 GeV primary all four models are suitable for the initial interaction, the evolution below the transition energy is modelled by UrQMD.

## 2 Event classification

Ground level observables are affected by the interplay of what happens in the first, highest energy, interaction and the subsequent lower energy processes. Here the first interaction will be characterised by the inelasticity $\kappa$ of the event and the type of the leading particle, i.e. the secondary with the highest energy. The inelasticity is defined as

$$\kappa = 1 - \frac{E_{\text{LP}}}{E_{\text{FI}}} \approx 1 - x_F^{\text{LP}}, \tag{1}$$

where $E_{\text{LP}}$ is the energy of the leading particle and $E_{\text{FI}}$ the total energy of all final state particles. This definition ensures that $\kappa$ is in the range $[0,1]$ and is insensitive to small violations of energy momentum conservation that are observed in all models. The inelasticity is related to Feyman's scaling variable $x_F$ and provides a qualitative measure for the amount of energy that goes into the production of new particles.

For the leading particle we differentiate between nucleons, the muonic family, which contains particles that either directly or via decays contribute to the ground level muon flux, the EM component, which leads to electromagnetic showers and others, which are less important for the ground level observables. Figure 2 shows the inelasticity distribution in EPOS-LHC for

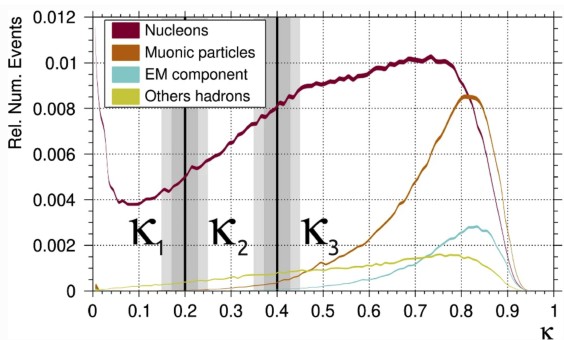

| Nucleons: | $p, n$ |
| Muonic family: | $\mu, \pi, K$ |
| EM component: | $\gamma, e$ |
| Others: | $\Lambda, \Sigma, \Xi, \Omega \dots$ |

Figure 2: Inelasticity distributions for events with different types of leading particles in EPOS-LHC, and definition of the particle families used to classify events. Antiparticles and different charge states are implied. The coloured bands in the plot visualise the statistical uncertainties of the simulation, the shaded regions indicate the systematic uncertainties in the definition of $\kappa$ due to event-by-event fluctuations in the amount of energy violation discussed in the appendix.

events with different types of leading particles. For small inelasticities the subsequent shower evolution is driven by leading nucleons, particles from the muonic family become important at large inelasticities. In the following we consider three regions: the elastic and diffractive region $\kappa_1 \in [0, 0.2]$, the transition region $\kappa_2 \in [0.2, 0.4]$, and the highly inelastic regime $\kappa_3 \in [0.4, 1]$.

# 3 Ground level observables

## 3.1 Lateral distributions of muons and electromagnetic energy flow

Figure 3 shows the lateral distributions of muons and the electromagnetic energy flow at ground level when the leading particle of the first interaction is a nucleon. As a function of the distance to the shower axis, the muon flux is shown in units of muons per square meter, for the electromagnetic component the energy deposit is given in GeV per square meter.

The electromagnetic energy flow is dominated by events from the diffractive region since in events with small inelasticity the bulk of the particle production happens deeper in the atmosphere, with the consequence that more of the electromagnetically interacting particles reach the ground. For the same reason also a sizeable fraction of the muon flux comes from the diffractive $\kappa_1$ region. In addition there is a large contribution from highly inelastic events, where many particles are created that decay into muons. The model with the largest number of muons close to the shower centre is QGSJetII-04. The excess is even more pronounced for events with a leading particle from the muonic family [6].

## 3.2 Muon flux at ground level

The overall numbers of muons at ground level are more similar since the area at small distances from the shower centre is only small fraction of the total. The average number of muons per event as a function of the inelasticity is shown in fig. 4, together with the $\kappa$ distributions of those events that dominate the muon flux at ground level, i.e. those with a leading nucleon or a leading particle from the muonic family. In the muon numbers the HE models show a discontinuity at $\kappa \approx 0.2$. Below that value the first two interactions, above only the first interaction is modelled by one of EPOS-LHC, QGSJetII-04 or SIBYLL 2.3c. The spread of the curves

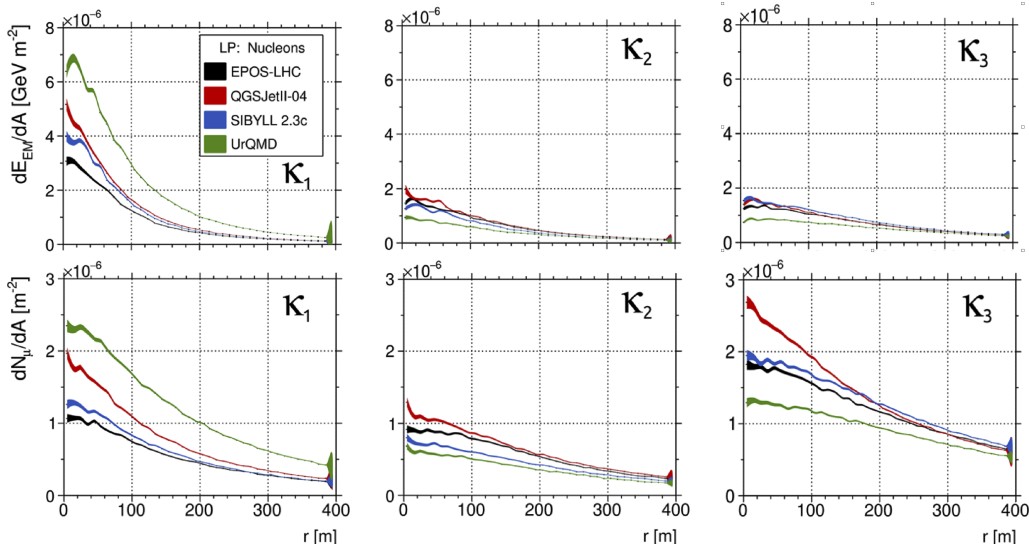

Figure 3: Lateral distributions electromagnetic energy flow (top row) and of muon numbers (bottom row) for different inelasticity regions and different HE interaction models.

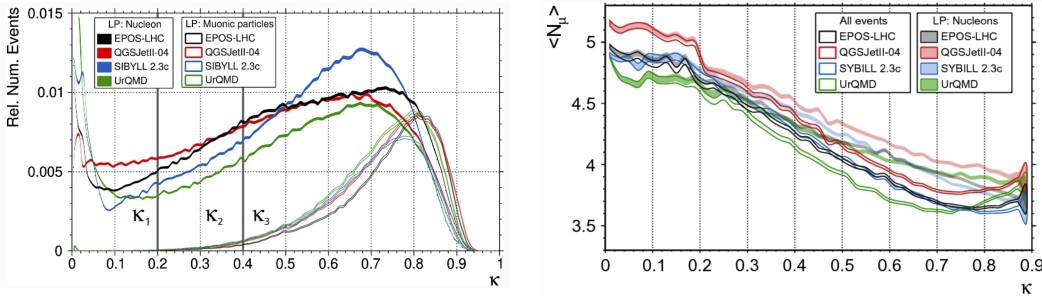

Figure 4: Inelasticity distributions (left) of the first interaction for events where the leading particle is a nucleon or belongs to the muonic family, and (right) average number of ground-level muons per event as a function of the inelasticity $\kappa$. The filled bands are for events with leading nucleons in the first interaction, the open ones are for all events.

for $\kappa > 0.2$ thus shows the importance of the first interaction for ground level observables, the jump at $\kappa \approx 0.2$ reflects the differences of the physics modelling between the HE models and UrQMD at the transition energy. The difference is largest for QGSJetII-04. Smaller but still significant jumps are observed for EPOS-LHC and SIBYLL 2.3c.

## 4 Conclusions

Studies of hadronic interaction models in the transition region between the HE and LE regime reveal significant differences in the prediction of ground level observables. Those can be traced to differences in the final states of the first interaction generated by the HE models and a discontinuity in the physics modelling when switching from a HE model to the LE model. Improvements of the models are expected from comparing their predictions to existing and upcoming data on inclusive particle production cross-sections from accelerator experiments. Here nucleon-nucleon centre-of-mass energies $\sqrt{s_{NN}}$ of $O(10)$ GeV are probed by

e.g. NA61/SHINE at the CERN-SPS, of $O(100)$ GeV by LHCb fixed target data and $O(10)$ TeV by proton-proton and proton-lead collisions recorded by the ALICE, ATLAS, CMS and LHCb experiments. Key data [12] for the understanding of cosmic-ray induced air showers is also expected from measurements of proton-oxygen and oxygen-oxygen collisions that are scheduled for Run 3 of the LHC.

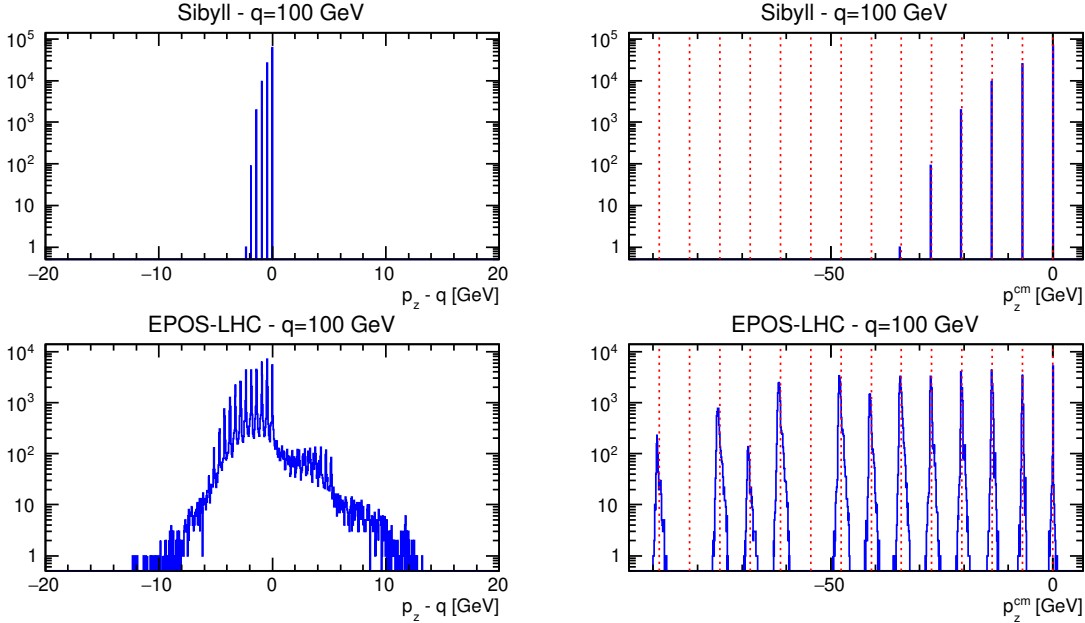

Figure 5: Violation of momentum conservation in SIBYLL 2.3C and EPOS-LHC. The left hand plots show the effect in the lab system, the right hand plots in the nucleon-nucleon centre-of-mass system. The red dashed lines indicate the expected values when $N = 1, 2, \ldots$ target nucleons are involved in the interaction.

## A  Energy-momentum violation in air shower models

Using the CRMC interface [13], the hadronic interaction models can be run standalone in order to compare to accelerator data or for simple checks of the kinematics. Taking for example an incoming proton with a lab momentum $q = 100$ GeV/$c$ colliding with a nucleus at rest, momentum conservation requires that the momentum sum of all final state particles is 100 GeV/$c$. As shown in fig. 5 there are deviations.

In SIBYLL-2.3c the effects are small, with slight offsets that are related to the number of target nucleons involved in the interaction. This can be seen by boosting the event to the nucleon-nucleon centre-of-mass, where the actual event generation happens. For an interaction with $N > 1$ target nucleons and a boost that goes to the centre-of-mass for $N = 1$, the boost generates a momentum excess of $1 - N$ times the centre-of-mass momentum of the incoming proton. For SIBYLL-2.3c one finds delta-functions that are marginally displaced from the expected values and correspond to the small shifts seen in the lab system.

For EPOS-LHC the centre-of-mass distributions show a sizeable smearing around the expected values, which correspond to violations of momentum conservation of up to 10 GeV/$c$ in the lab system. The large deviations can be avoided by improving the numerical precision of

the model, changing in line 17 of the function `epos-uti.f` the value $0.5$ to e.g.$0.005$ [14]:

```
if(iLHC.eq.1) errlim=max(0.00005,0.5/engy)
```

The boost to the nucleon-nucleon centre-of-mass allows one also to determine the number of interacting target nucleons. Here, in SIBYLL-2.3c only up to 5 target nucleons interact whereas in EPOS-LHC the full range from 1–14 except for N=9 and N=13 is covered.

The importance of correctly modelling sub-TeV hadronic interactions for air showers is underlined by fig. 6, which, for 1 PeV primaries, shows the average number of interactions in the shower as a function of the particle energy. The number of interactions rises exponentially as the particle energy drops, such that the bulk of the particle production occurs at energies at or below the transition energy. The plot also shows that the number of sub-TeV interactions in SIBYLL 2.3c and QGSJetII-04 is up to 20% lower than for EPOS-LHC.

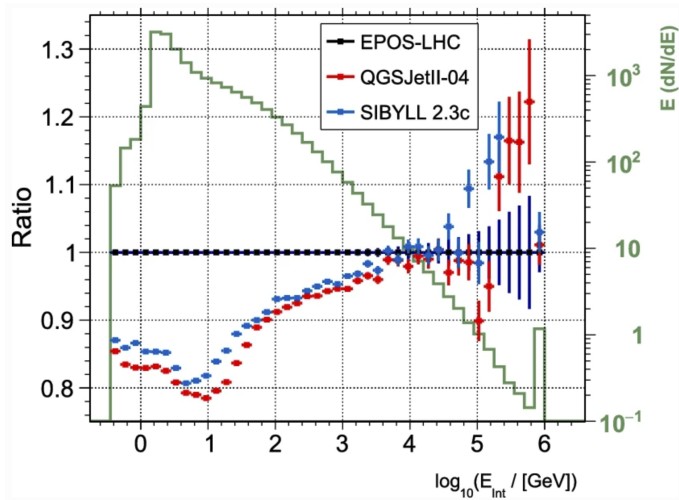

Figure 6: Average number of interactions as a function of the particle energy in air showers initiated by 1 PeV protons hitting the atmosphere. The histogram (green, right hand ordinate) is the prediction by EPOS-LHC, the points (left hand ordinate) show the ratios to EPOS-LHC for SIBYLL 2.3c and QGSJetII-04.

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
