# Peer review of "Sub-TeV hadronic interaction model differences and their impact on air showers"

_SciPost Physics Proceedings, doi:SciPost Phys. Proc. 15, 015 (2024)_

## Round 1 · Referee Report · Anonymous (Referee 1) · 2022-11-25

Strengths

The submission is clearly written and discusses an interesting issue that was previously overlooked and which is somewhat surprising. Results of air shower simulations vary strongly in the sub-TeV region when hadronic interactions models are essentially only used for the first interaction.

Many publications focus on the high-energy behavior of these models, when they are extrapolated beyond the energy covered by collider and fixed-target experiments. The energy range discussed here is within the reach of these experiments. The large variation in the model response either indicates that the models are not well tuned to data at sub-TeV energies or that the available experimental data does not cover the phase-space in which these models differ.

Weaknesses

The introduction lacks a survey of other works that have studied discrepancies between models in the high-energy region. The authors only cite one of their own papers. The introduction should give context by citing also other works and contrast the present work to them.

While the findings are very interesting, the conclusions are not very strong. What specific measurements are needed and could be performed at colliders to validate the models? Do we have the required data and it is merely a problem of tuning or is the data missing? Are new experiments needed? The study presents the results as a function of the unusual variable kappa, which is not commonly used in collider experiments. Can kappa be measured in collider experiments?

Report

The paper is of high-quality and acceptance criteria are met, however, I kindly ask for a minor revision to expand the introduction and conclusions.

Requested changes

The authors should address the two weaknesses in the introduction and conclusion of the paper. In addition, I suggest the following minor improvements:

p 1 - “Model differences in the description of high energy interactions are discussed in [1], the low energy region has been studied in [2], some key results of which are summarised below.” Please use numerical energy ranges here, “high-energy” and “low-energy” mean different things to different people. There are many papers that study model differences in high-energy interactions in addition to ref 1. The authors should give a fair sample of the existing literature in addition to citing their own paper.

p 2 - Fig 2. The shaded regions around the vertical lines are not explained. If these do not have any meaning, they should be removed. I guess they indicate that the numerical choices for the thresholds between the three regions is somewhat arbitrary, but this can be said in the text. The meaning of the bands around the squiggly graphs is also not explained in the caption. I assume it is the statistical uncertainty, that should be clarified in the caption.

p 3 - “κ1 ∈ [0, 0.2], the elastic and diffractive region…” The structure in this sentence is not regular, which makes it harder to read. A regular pattern is better: “… three regions: the elastic and diffractive region K1 in [0, 0.2], the transition region K2 in [0.2, 0.4], and the highly inelastic regime K in [0.4, 1].” This construction also reduces the amount of commas.

---

## Round 2 · Author Response

We thank the reviewer for the careful reading and the constructive feedback on the manuscript. The original version was a bit on terse side also because we wanted to stick to the target page limit, i.e. we are happy for the opportunity to expand the text a bit.

---

## Round 2 · List of Changes

The revised version

  • implements the suggested improvement on the wording
  • has an expanded introduction which puts the subject into broader context and cites additional paper
  • has more explanations in the caption of figure 2
  • puts, in the conclusions, more emphasis on comparisons to accelerator data. We did not enter into a discussion of whether kappa is a good variable for accelerator experiments, but simply point out that information from that source will mainly be inclusive particle production cross sections.

---

## Editorial Decision

published